# Wheat Sensitivity and Functional Dyspepsia: A Pilot, Double-Blind, Randomized, Placebo-Controlled Dietary Crossover Trial with Novel Challenge Protocol

**DOI:** 10.3390/nu12071947

**Published:** 2020-06-30

**Authors:** Michael D. E. Potter, Kerith Duncanson, Michael P. Jones, Marjorie M. Walker, Simon Keely, Nicholas J. Talley

**Affiliations:** 1Faculty of Health and Medicine, University of Newcastle, Callaghan, NSW 2308, Australia; michael.potter@newcastle.edu.au (M.D.E.P.); mike.jones@mq.edu.au (M.P.J.); marjorie.walker@newcastle.edu.au (M.M.W.); simon.keely@newcastle.edu.au (S.K.); nicholas.talley@newcastle.edu.au (N.J.T); 2Australian Gastrointestinal Research Alliance (AGIRA), Hunter Medical Research Institute, New Lambton Heights, NSW 2305, Australia; 3Department of Gastroenterology, John Hunter Hospital, New Lambton Heights, NSW 2305, Australia; 4Psychology Department, Macquarie University, Macquarie Park, Sydney, NSW 2109, Australia

**Keywords:** non coeliac wheat sensitivity, gluten, FODMAPs, functional dyspepsia

## Abstract

Introduction: Functional dyspepsia (FD), characterised by symptoms of epigastric pain or early satiety and post prandial distress, has been associated with duodenal eosinophilia, raising the possibility that it is driven by an environmental allergen. Non-coeliac gluten or wheat sensitivity (NCG/WS) has also been associated with both dyspeptic symptoms and duodenal eosinophilia, suggesting an overlap between these two conditions. The aim of this study was to evaluate the role of wheat (specifically gluten and fructans) in symptom reduction in participants with FD in a pilot randomized double-blind, placebo controlled, dietary crossover trial. Methods: Patients with Rome III criteria FD were recruited from a single tertiary centre in Newcastle, Australia. All were individually counselled on a diet low in both gluten and fermentable oligo-, di-, mono-saccharides, and polyols (FODMAPs) by a clinical dietitian, which was followed for four weeks (elimination diet phase). Those who had a ≥30% response to the run-in diet, as measured by the Nepean Dyspepsia Index, were then re-challenged with ‘muesli’ bars containing either gluten, fructan, or placebo in randomised order. Those with symptoms which significantly reduced during the elimination diet, but reliably reappeared (a mean change in overall dyspeptic symptoms of ≥30%) with gluten or fructan re-challenge were deemed to have wheat induced FD. Results: Eleven participants were enrolled in the study (75% female, mean age 43 years). Of the initial cohort, nine participants completed the elimination diet phase of whom four qualified for the rechallenge phase. The gluten-free, low FODMAP diet led to an overall (albeit non-significant) improvement in symptoms of functional dyspepsia in the diet elimination phase (mean NDI symptom score 71.2 vs. 47.1, *p* = 0.087). A specific food trigger could not be reliably demonstrated. Conclusions: Although a gluten-free, low-FODMAP diet led to a modest overall reduction in symptoms in this cohort of FD patients, a specific trigger could not be identified. The modified Salerno criteria for NCG/WS identification trialled in this dietary rechallenge protocol was fit-for-purpose. However, larger trials are required to determine whether particular components of wheat induce symptoms in functional dyspepsia.

## 1. Introduction

Functional dyspepsia (FD) is a troublesome gastrointestinal disorder that affects the health and wellbeing of more than 15% of the population [1]. It is characterised by symptoms referrable to the gastroduodenal region of the abdomen, including early satiety, post-prandial fullness, and epigastric pain [2]. FD overlaps with irritable bowel syndrome (IBS) and may be mislabelled as such. Both disorders are associated with meal-related symptoms [3]. People with FD often report that certain dietary triggers exacerbate symptoms, with wheat and/or gluten commonly implicated [4]. Functional dyspepsia is also closely linked with wheat sensitivity in epidemiological studies [1].

Duodenal eosinophilia has been observed in biopsy tissue samples obtained at upper endoscopy from patients with FD, particularly in those with post-prandial distress syndrome [5,6,7]. In a case-control study from Sweden of 51 adults with FD, duodenal eosinophilia was significantly increased in cases compared with controls, with a mean of 33.1 and 34.6 eosinophils per five high-power fields (HPF) in FD cases in the first and second part of the duodenum (D1 and D2), compared with 18.4 and 18.6 in controls [5]. A study of 33 patients from an Australian centre replicated these observations, demonstrating duodenal (D2) eosinophilia in patients with the post-prandial distress subtype of FD [6]. Mechanistic studies have implicated duodenal eosinophils with impaired intestinal duodenal barrier and neuronal functioning [8,9], pointing towards an underlying mechanism for the disorder. An allergen (potentially wheat) or infection may lead to barrier disruption and the generation of a Th2 or Th17 type immune response, which induces recruitment and degranulation of eosinophils, affecting the submucosal nervous system and altering gastroduodenal function [10]. In the absence of a validated biomarker, it is not currently possible to attribute FD symptom generation to a specific component of wheat [11,12].

Wheat is also commonly implicated in unexplained gastrointestinal and extraintestinal symptoms termed ‘non-coeliac gluten or wheat sensitivity’ (NCG/WS). This disorder is characterised by wheat sensitivity (specifically self-reported adverse physiological symptoms after wheat ingestion) in the absence of demonstrable wheat allergy or coeliac disease [13,14]. Symptoms commonly reported include those which overlap with IBS, such as bloating, constipation, abdominal pain, and diarrhoea. Dyspeptic symptoms also affect more than half of those with NCWS [15] and extraintestinal symptoms are common [14]. Similar to FD, duodenal eosinophilia has been reported in NCG/WS [7]. In a group of 276 patients with NCG/WS, diagnosed by a double-blind, placebo-controlled challenge, the duodenal eosinophil count was measured to be significantly higher than both coeliac and IBS controls (63 per 10 HPF vs. 38 and 31 respectively, *p* < 0.0005) [7]. Current consensus criteria requires a blinded placebo controlled crossover challenge of gluten in order to make the diagnosis of non-coeliac gluten sensitivity (NCGS) [16].

Fructans are a type of fermentable carbohydrate common in wheat. They are implicated in IBS symptoms such as bloating and changes in bowel habit [17], but less is known about their contribution to other gastrointestinal symptoms, including dyspepsia. Wheat contributes at least 70% of total dietary fructans in the United States, averaging 2.6 g of inulin-type fructans and 2.5 g of fructo-oligosaccharide per day [18]. A food item is considered to be a substantial food sources of fructans if it contains more than 0.5 g of fructans per serving [19]. Wheat also contains amylase trypsin inhibitors (ATIs), which are non-gluten proteins capable of activating the innate immune system via interaction with toll-like receptor [14].

To better understand the relationship between wheat and symptom induction in people with FD, the effects of gluten, fructan- type FODMAPs (fermentable oligo- di- mono- saccharides and polyols) and ATIs need to be differentiated. This is possible if the consensus protocol proposed for NCGS [16] is extended to account for fructans and ATIs. The Salerno Experts’ Criteria for diagnosis of NCGS (2015) recommends a double-blind, randomised crossover trial that provides at least eight grams of gluten (cooked into food, not capsule) a day in the gluten challenge phase. This dose approximates the average daily intake of gluten in Western countries (10–15 g) [20]. Adding a wheat fructan dietary challenge arm to the Salerno criteria protocol would mean the role of gluten and fructans could be differentiated. A daily dose of fructans of at least 2.5 g would be expected to induce symptoms in those with fructan sensitivity, but not in the general population.

The primary aim of the current study was to ascertain whether a wheat free diet, specifically a gluten-free and low-FODMAP diet, induced a significant reduction in symptoms in patients with functional dyspepsia. Our secondary aim was to demonstrate whether gluten or fructans were responsible for the improvement in those who responded to the wheat free diet, using a placebo-controlled, blinded re-challenge. We also tested a novel double-blind, randomised crossover trial design for gluten and fructan challenges to differentiate potential wheat components implicated in FD symptom induction.

## 2. Methods

### 2.1. Participants

Adult participants (age 8–80 years) were recruited from the outpatient gastroenterology clinics at John Hunter Hospital, a tertiary referral centre located on the mid-north coast of New South Wales, Australia (Appendix A). Participants fulfilled Rome III criteria for functional dyspepsia based on symptoms and a negative upper endoscopy [20]. All participants tested negative for coeliac disease (with negative anti-tissue transglutaminase IgA and normal duodenal biopsies) and wheat allergy (negative wheat specific serum IgE). Those with inflammatory bowel disease, active malignancy or infection, and pregnant patients were excluded.

### 2.2. Protocol

The study design was modified from the Salerno criteria, the currently accepted standard for the diagnosis of NCG/WS [16]. Participants, ideally on a normal wheat-containing diet for 4 weeks, completed a food frequency questionnaire [21], and were instructed on a low-FODMAP, gluten-free diet by a clinical dietitian (Figure 1).

Symptoms were assessed at baseline and after 4 weeks using the validated Nepean Dyspepsia Index (NDI) [22]. A food frequency questionnaire was used to calculate the change (mean, grams per day) in FODMAP intake over the run-in diet period [21], and a validated questionnaire was used to ensure compliance with the gluten-free aspect of the diet [23] (Appendix A). Those with a significant reduction in symptoms after the run-in diet (defined as ≥30% reduction in NDI score) were eligible for the rechallenge phase. This involved continuation of the low-FODMAP, gluten-free baseline diet with the addition of one ‘challenge’ bar per day (to replace a snack) for one week at a time, separated by a week-long washout period. The order in which the three ‘challenge’ bars were consumed by participants was randomized using a computer generated randomization algorithm, with the bar order contained in written instructions that were stored in a sealed envelope and given to each participant at the beginning of the re-challenge phase, blinding the research assistant to treatment allocation. Participants were assigned bars containing fructans (approximately 6.9 g inulin, without gluten), gluten (approximately 8.5 g gluten [24], with low fructan content), and placebo (without fructans or other FODMAPs, gluten-free ingredients) (Appendix A). The bars were independently tested for FODMAP content by an external laboratory prior to trial commencement (FODMAP friendly, Pty Ltd., Victoria, Australia) to confirm FODMAP content. All bars were nutritionally equivalent and indistinguishable in look, texture, and taste (Figure 2) but labelled as Bar A, Bar B, or Bar C to align with double-blinded randomization process.

Dyspeptic symptoms were measured daily during the rechallenge phase and weekly during the washout weeks using a numbered visual analogue scale (VAS) (3 main symptoms; post-prandial fullness, epigastric pain, early satiety, each scored 0–10) [16].

This study received Hunter New England Human Research Ethics Committee approval on March 18, 2018 (ethics approval number: 2019/ETH01181). The study is registered as Australia New Zealand Clinical Trial: ID Number: 380018.

### 2.3. Sample Size and Statistics

Our sample size calculation was based on the hypothesis that a dietary trigger is responsible for symptoms with FD, and that the dietary response would be due to gluten or FODMAP intake. Using repeated measures analysis of variance, with a power of 0.8 and a significance of 0.025 (0.05/2 for dual hypothesis) and a delta of 0.5, we calculated that we required 41 subjects to enter the dietary challenge phase of the trial. Assuming that 30% of subjects would not respond to the run-in diet and be eliminated, we estimated 58 participants would need to commence the study. Statistical analysis was performed using STATA software (StataCorp, TX, USA).

Changes in FODMAP intake and symptoms between run-in and diet phase were evaluated via repeated measures analysis of variance. Association between baseline factors and change in symptoms was evaluated via linear regression.

## 3. Results

Eleven participants were enrolled in the study between July 2018 and February 2019 (75% female, mean age 43 years). Regarding the functional dyspepsia subtype, four had epigastric pain syndrome, two had postprandial distress, and five fulfilled criteria for both (overlap syndrome). Four participants were already following a partial exclusionary diet (three partially avoiding gluten and/or FODMAPs, one following a low FODMAP diet). The sample size of 41 participants was not achieved in the study timeframe due to logistical reasons.

After the run-in diet, the mean FODMAP intake decreased from 40.1 g to 17.1 g (*p* = 0.14) (Figure 3 and Appendix A), and all were adherent to gluten exclusion based on the applied questionnaire [23].

Of the initial 11 participant cohort, nine completed the run-in diet phase. The gluten-free, low-FODMAP run-in diet led to an overall improvement in symptoms of functional dyspepsia (mean NDI symptom score 71.2 vs. 47.1, *p* = 0.087; Figure 4 and Appendix A).

There was no significant association between the baseline eosinophil count and the magnitude of change in dyspeptic symptoms during the run-in diet period (*p* = 0.45, linear regression, Figure 5). Two participants reported worsening of constipation symptoms, and one worsening of bloating and constipation requiring cessation of the run-in diet.

## 4. Discussion

An overall reduction in dyspeptic symptoms was observed with removal of wheat gluten and fructans, but the association was not significant. Therefore, we have not demonstrated a link between FD and NCG/WS. Early termination of the study due to under-recruitment and low eligibility for the challenge phase resulted in the dietary re-challenge not being conducted by a meaningful number of participants.

Our study demonstrated a trend towards improvement in dyspeptic symptoms on a wheat-free, low-FODMAP diet. The mean total FODMAP intake reported at baseline in this FD cohort was higher than total FODMAP intake of IBS sufferers who had returned to a ‘habitual’ diet (40.1 versus 29.4 g per day) in a study on long-term outcomes of a FODMAP-modified diet, using the same dietary assessment tool [25]. The post ‘run-in’ FODMAP intake in our study approximated the intake of those who continued on a FODMAP restricted diet in the same long-term outcome study (17.1 versus 20.6 g per day) [25]. Despite the low participant numbers at both baseline (n = 8) and post ‘run in’ (n = 5), these results indicate that a low FODMAP intake was achieved in the ‘run-in’ phase by this FD cohort.

After four weeks of a gluten-free diet and 57% reduced mean FODMAPs intake (grams per day), participants (n = 9) reported a 33% reduction in FD symptom score. This is consistent with previous observational studies [4] and one intervention study [26] that demonstrated a link between wheat-based foods and symptoms in people with functional dyspepsia. Elli et al. [26] recruited 134 participants with a Rome III diagnosis of FD and IBS. Seventy-five percent (n = 98) reported improvement in symptoms on an initial GFD and progressed to the blinded trial phase. Of this participant subset, 14% (10% of original study sample) reported recurrence of symptoms with blinded gluten capsule challenges, fulfilling a clinical diagnosis of NCG/WS. Two out of four specific symptoms that showed a significant association with gluten ingestion in the blinded challenge are associated with FD rather than IBS (post prandial fullness (*p* = 0.01), and early satiety (*p* = 0.03)) [26].

We did not demonstrate an association between the number of duodenal eosinophils at baseline and the subsequent symptom response, although given the small numbers of participants recruited this may be due to under-powering. Further studies with a sample size of at least 50 participants are needed to establish whether duodenal eosinophilia may be a biomarker for wheat sensitive FD.

### 4.1. Limitations

Only 11 participants enrolled in this trial from an expected sample size of approximately 60 people. It is possible that the dietary requirements of the trial were prohibitive for prospective participants, or that the study needed a longer recruitment period or more study sites. Twenty-seven percent (n = 3/11) of participants completed the 14-week study. This high attrition rate was partly attributable to participants being ineligible to proceed into the crossover trial if they did not achieve a > 30% reduction in their NDI score. In future studies, our study criteria will be revised to account for, but not exclude based on symptom reduction.

As participants who experienced exacerbation of symptoms did not complete the post run-in diet FFQ, it is not known whether they had reduced FODMAP intake, or if other lifestyle factors, such as stress, influenced symptom induction. The study was under-powered to demonstrate an association between the number of duodenal eosinophils at baseline and the subsequent symptom response after the ‘run-in’ diet. It was also under-powered to detect a signal that one type of bar caused symptom worsening compared to the other two bars. Finally, ATIs from wheat have been shown to activate TLR4, triggering innate immune effects and increasing low level inflammation [14]. ATIs were not measured in this study but should be considered in future studies that aim to distinguish between respective wheat components in NCG/WS and FD.

Following a gluten-free, low-FODMAP diet is expensive and difficult to maintain, even if the individual receives advice and support from a dietitian and experiences a reduction in symptoms. We recommend that future studies involve the provision of some low FODMAP, gluten-free grocery items to reduce the financial burden, and to increase participant retention and dietary compliance.

### 4.2. Implication for Practice and Research

Despite this study being underpowered to detect an association between wheat component intake and FD symptom induction, there are some implications for clinicians and researchers.

We emphasise the importance of using Rome criteria for diagnosis of FD, and differentiation from IBS (or diagnosis of overlapping FD/IBS) as a basis for studies investigating the role of wheat and other potential symptom triggers.

Although not fully elucidated in this study, we recommend that clinicians consider the potential involvement of wheat in FD symptom induction and management. Dietitians are well positioned to support people with FD to maintain an exclusion diet that balances symptom management, diet variety, and nutritional adequacy.

We recommend that future research studies allocate funding for provision of gluten-free, low-FODMAP groceries to participants to increase recruitment, retention, and dietary compliance. We expect that assessing symptom reduction in the ‘run-in’ phase (without setting a strict exclusion cut-off) will increase retention into the crossover trial phase of FD and NCG/WS studies. Additional investment of research resources towards understanding the respective roles of wheat gluten, fructans, and ATIs in the intestinal and extraintestinal symptoms and characteristics of FD will substantially progress our understanding and management of functional dyspepsia.

## 5. Conclusions

Although a gluten-free, low-FODMAP diet led to a modest overall reduction in symptoms in this cohort of FD patients, this was not significant, and a specific trigger could not be identified using this dietary rechallenge protocol. Further larger studies are required to explore whether a wheat-free diet (either a gluten-free or low-FODMAP diet) may be of use in treating functional dyspepsia. Wheat may contribute, but even if it is a factor in FD pathogenesis, it does not seem to be solely responsible for symptoms.

## 6. Patents

Patented functional grain product concepts used in low-FODMAP, high-gluten muesli bar development (Australian Patent No. 2014262285; New Zealand Patent No. 629207; South Africa Patent No. 2015/07891).

## Figures and Tables

**Figure 1 nutrients-12-01947-f001:**
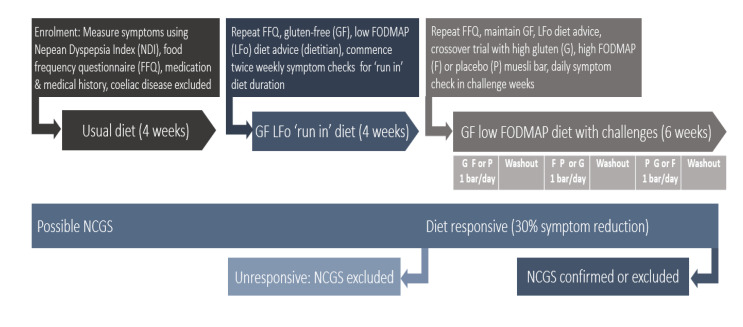
Non coeliac gluten/wheat sensitivity protocol overview including: ‘Run in phase’ (dark grey); baseline clinical testing, baseline diet assessment; gluten-free, low-fermentable oligo-, di-, mono-saccharides, and polyols (FODMAP) diet instruction by dietitian, four-week ‘run-in’ diet phase with pre-post symptom measurement, and second-daily dyspeptic symptom measurement; ‘Dietary challenge phase’ (dark blue); gluten-free, low-FODMAP diet continued, one challenge bar per day for one week (order of bars randomized) with high gluten, high fructan/FODMAP and placebo bars, with symptoms measured daily using a visual, analogue scale. Key: GF: Gluten free; FODMAP: Fermentable oligo- di- mono- saccharides and polyols; NCGS: Non-coeliac gluten sensitivity.

**Figure 2 nutrients-12-01947-f002:**
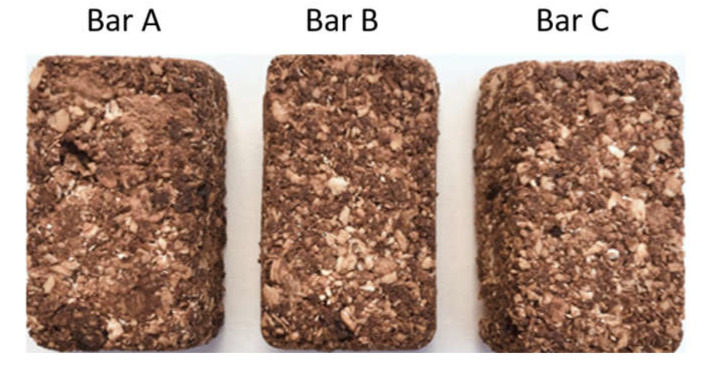
Three visually indistinguishable, nutritionally equivalent ‘muesli’ bars with differing gluten and fructan contents used for a functional dyspepsia randomized, double-blind crossover trial.

**Figure 3 nutrients-12-01947-f003:**
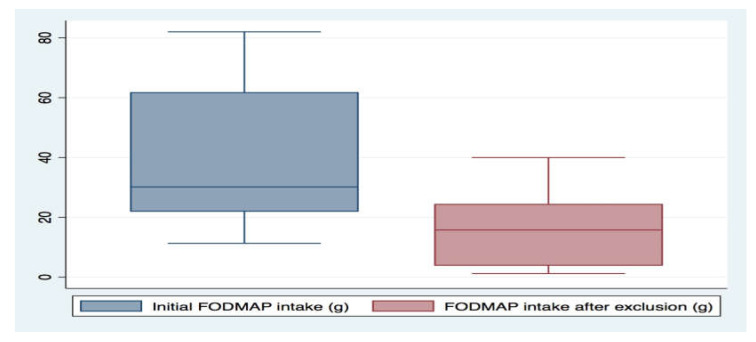
Total FODMAP (fermentable oligo-, di-, and monosaccharides, and polyols, gram per day) intake before (n = 8) and after (n = 5) commencement of low-FODMAP, gluten-free diet, as measured by the food frequency questionnaire (*p* = 0.14, Wilcoxon signed-rank test).

**Figure 4 nutrients-12-01947-f004:**
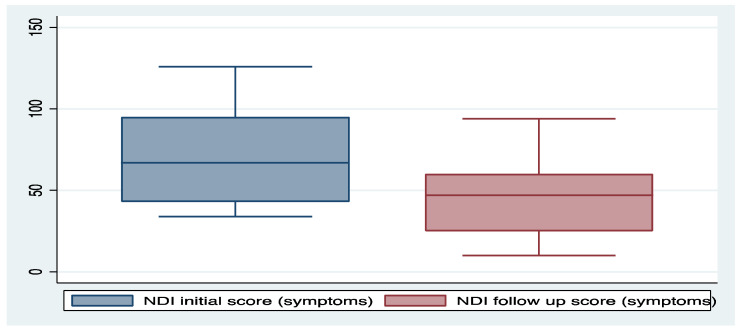
Mean symptom scores before (n = 10) and after (n = 8) the gluten-free, low-FODMAP diet (*p* = 0.087, Wilcoxon sign rank test). NDI—Nepean dyspepsia index.

**Figure 5 nutrients-12-01947-f005:**
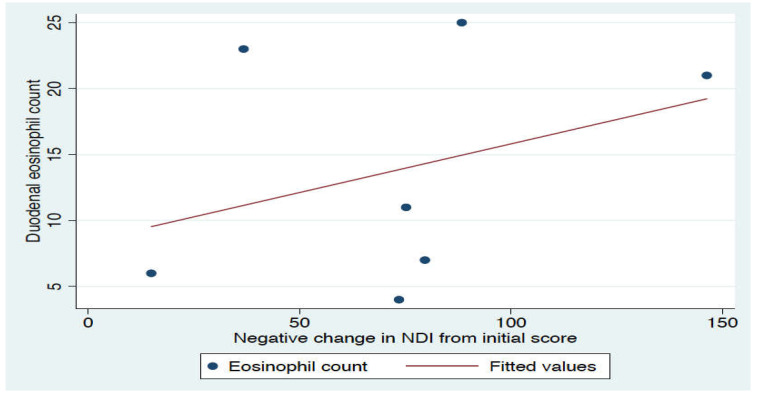
Relationship between duodenal eosinophil count and magnitude of change in the severity of dyspeptic symptoms measured with the Nepean Dyspepsia Index (NDI) after the run-in diet (*p* = 0.45, r^2^ = 0.12). Duodenal eosinophil count expressed per mm^2^. Four subjects qualified for the rechallenge phase, based on a >30% reduction in their NDI score. Three out of four of these participants completed the protocol. Meaningful analysis of the data was not possible, however there was no signal that one bar caused symptom worsening compared to other bars.

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
