# Peer review of "Wheat Sensitivity and Functional Dyspepsia: A Pilot, Double-Blind, Randomized, Placebo-Controlled Dietary Crossover Trial with Novel Challenge Protocol"

_nutrients, 2020, doi:10.3390/nu12071947_

Round 1

Reviewer 1 Report

Great design  you already know this but more numbers are needed to make this significant. 

Author Response

Comment: Great design  you already know this but more numbers are needed to make this significant. 

Response to reviewer 1:

Thanks you for your encouraging comments about the study design. We agree that the study was underpowered, but that it is important to report methods and preliminary findings in this important field of research.

Reviewer 2 Report

It is a well designed study. Aims and method are clearly described but, even it’s a pilot study, the enrolled patients is really very few

Abstract

 “All were individually counselled on a diet low in both gluten and fermentable oligo-, di-, mono-saccharides and polyols (FODMAPs)..” but LowFODMAP diet is naturally a low gluten diet. The authors should clarify this topic because, on the contrary, in Methods they deal with “ a low-FODMAP, gluten free diet..” The latter looks like more correct.

Results

“9/11 participants completed the elimination diet phase of whom 4 qualified for the rechallenge phase.” Even for a pilot study the amount of patients involved in this study seems very little.

Discussion

“We emphasise the importance of using Rome criteria for diagnosis of FD, and differentiation from IBS (or diagnosis of overlapping FD/IBS) as a basis for studies investigating the role of wheat and other potential symptom triggers” Did the authors study the presence of IBS symptoms in their patients, in order to detect the overlap between FD/IBS and the possible effect of the LFD and of the rechallenge with gluten, fructans and placebo? Otherwise this statement is not justified.

The conclusions and the suggestions the authors provide are completely sharable but, because of the very low number of patients, they are based on hypotheses rather than the study results

Author Response

Reviewer 2

It is a well-designed study. Aims and method are clearly described but, even it’s a pilot study, the enrolled patients is really very few

Response: Thank you for your positive feedback on the study design. We address the comment about the participant numbers in our response to your specific comment (and those from other reviewers) about this below (under Results).

Comment: Abstract

 “All were individually counselled on a diet low in both gluten and fermentable oligo-, di-, mono-saccharides and polyols (FODMAPs)..” but a Low FODMAP diet is naturally a low gluten diet. The authors should clarify this topic because, on the contrary, in Methods they deal with “ a low-FODMAP, gluten free diet..” The latter looks like more correct.

Response:

Thank you for your comment the discrepancy between how the diet is described in the abstract and methods. The abstract had been amended as follows:

All were individually counselled on a diet low fermentable oligo-, di-, mono-saccharides and polyols (FODMAPs), gluten-free diet by a clinical dietitian……..

Comment: Results

“9/11 participants completed the elimination diet phase of whom 4 qualified for the rechallenge phase.” Even for a pilot study the amount of patients involved in this study seems very little.

Response:

We recognise the very low participant numbers in this study and feel this has been fully acknowledged in the manuscript. Therefore. our aim for this manuscript is to facilitate advancement in the field by sharing the rigorous, detailed methodology and recommendations for future NCWS studies, which require the combination of high level clinical, dietetic and laboratory expertise.

Comment: Discussion

“We emphasise the importance of using Rome criteria for diagnosis of FD, and differentiation from IBS (or diagnosis of overlapping FD/IBS) as a basis for studies investigating the role of wheat and other potential symptom triggers” Did the authors study the presence of IBS symptoms in their patients, in order to detect the overlap between FD/IBS and the possible effect of the LFD and of the rechallenge with gluten, fructans and placebo? Otherwise this statement is not justified.

The conclusions and the suggestions the authors provide are completely sharable but, because of the very low number of patients, they are based on hypotheses rather than the study results.

Response:

Thank you for pointing this out. To emphasise that the Implications section is based on the additional methodological and preliminary data from this very small study, in addition to the existing evidence, section 4.2 has been changed as follows:

4.2. Implications for practice and research

Despite this study being underpowered to detect an association between wheat component intake and FD symptom induction, the addition of our findings and study methodology to existing evidence have some implications for clinicians and researchers.

Based on the existing evidence-base, combined with the methodological challenges of implementing blinded dietary challenge trials, we emphasise the importance of Rome criteria diagnosis of FD, and differentiation from IBS (or diagnosis of overlapping FD/IBS), as a basis for studies investigating the role of wheat and other potential symptom triggers. Repeated assessment of FD and IBS symptoms throughout a study (eg. Appendix B), and analysis of these in relation to dietary intake and the intervention are recommended.  

Reviewer 3 Report

I have no recommended amendments. While poor enrollment prevented this study from answering the primary questions, this manuscript has merit as a methodology paper. The protocol addresses a very important question that is key in understanding the interaction between food and functional dyspepsia. It presets a well developed plan/methodology for answering the question and provides valuable information on enrollment barriers and necessary sample size.

Author Response

Thank you for recognising the value in the presentation of the methodology of this paper. We hope that the amendments made in response to reviewer comments have further enhanced the value of this paper.

Reviewer 4 Report

Functional dyspepsia (FD) is a chronic condition characterized by upper digestive symptoms with no obvious cause. FD patients commonly report dietary triggers that exacerbate symptoms, including wheat and/or gluten. The authors attempted to probe the relationship between diet and FD symptoms using defined diets. The study aimed to ascertain the following points: (1) whether a wheat-free diet reduced FD symptoms; and (2) whether gluten or fructans improved patients who responded to the wheat-free diet. Based on a power-analysis, the authors estimated that 58 participants would be required to address these questions. Unfortunately, the study only enrolled 11 patients; far below the required patient number. Based on the small n, no significant differences were observed between FODMAP diet, FD symptoms, or eosinophil count. However, trends were observed and these findings may inform future studies. The authors were very straight forward with the intent of the study and the limitations and the discussion is very comprehensive. Despite the small n and obvious limitation of this study, this information is still important.

Comments:

  1. Line 74: Please change irritable bowel syndrome to IBS
  2. Line 100: In the methods, is there a misspelling in the ages? Is 8 supposed to be 18 (to be classified as an adult).
  3. For the graphs, could the individual values (per person) be graphed inside bar graph instead? It would be more visually informative.

Author Response

  1. Line 74: Please change irritable bowel syndrome to IBS

Response: Amended as suggested to IBS, and checked throughout. 

  1. Line 100: In the methods, is there a misspelling in the ages? Is 8 supposed to be 18 (to be classified as an adult).

Response: Thank you for picking this up. This error has been amended.

  1. For the graphs, could the individual values (per person) be graphed inside bar graph instead? It would be more visually informative.

Response: Thank you. The individual participant values have been included as Supplementary files.

See text line 166: After the run-in diet, the mean FODMAP intake decreased from 40.1g to 17.1g (p=0.14) (Figure 3 and Figure s3),

See text line 173:  The gluten free, low FODMAP run-in diet led to an overall improvement in symptoms of functional dyspepsia (mean NDI symptom score 71.2 vs 47.1, p=0.087; Figure 4 and Figure s4).

Figure s3: Individual participant fermentable oligo, di, mono-saccharide and polyol (FODMAP) intake pre and post 4 week ‘run in’ gluten-free, low FODMAP diet

Figure s4: Individual participant Nepean Dyspepsia Index (NDI) scores pre and post 4 week ‘run in’ gluten-free, low FODMAP diet  

Round 2

Reviewer 2 Report

I think the authors answered my previous comments and questions

No further comments